# Laser Microdissection-Mediated Isolation of Butterfly Wing Tissue for Spatial Transcriptomics

**DOI:** 10.3390/mps5040067

**Published:** 2022-08-11

**Authors:** Tirtha Das Banerjee, Shen Tian, Antόnia Monteiro

**Affiliations:** 1Department of Biological Sciences, National University of Singapore, Singapore 117557, Singapore; 2Science Division, Yale-NUS College, National University of Singapore, Singapore 138609, Singapore

**Keywords:** laser microdissection, *Bicyclus anynana*, spatial transcriptomics, RNAseq, eyespots, wing sectors

## Abstract

The assignment of specific patterns of gene expression to specific cells in a complex tissue facilitates the connection between genotype and phenotype. Single-cell sequencing of whole tissues produces single-cell transcript resolution but lacks the spatial information of the derivation of each cell, whereas techniques such as multiplex FISH localize transcripts to specific cells in a tissue but require a priori information of the target transcripts to examine. Laser dissection of tissues followed by transcriptome analysis is an efficient and cost-effective technique that provides both unbiased gene expression discovery together with spatial information. Here, we detail a laser dissection protocol for total RNA extraction from butterfly larval and pupal wing tissues, without the need of paraffin embedding or the use of a microtome, that could be useful to researchers interested in the transcriptome of specific areas of the wing during development. This protocol can bypass difficulties in extracting high quality RNA from thick fixed tissues for sequencing applications.

## 1. Introduction

Recent advances in spatial transcriptomics approaches have paved a new frontier in our understanding of tissue and organ development. The study of spatial transcriptomics using multiplex fluorescent in situ hybridization (multiplex-FISH) based techniques such as MERFISH [1], seqFISH [2], seqFISH+ [3], split-FISH [4] and osmFISH [5]; in situ RNA sequencing based techniques such as Slide-seq [6], and STARmap [7]; and single-cell RNA-seq followed by the deciphering of a cell’s spatial location [8,9,10,11,12] is revolutionizing biology at the molecular level. These techniques, however, all have limitations in terms of cost, time, and labor. Techniques such as MERFISH, seqFISH+, and split-FISH require thousands to several thousands of primary probes and different fluorescently labeled secondary probes for the detection of mRNA species [2,3,4], in situ sequencing based techniques require complex preparatory steps and imaging setups [6,7], and single-cell RNAseq requires dissociation of cells from the tissue resulting in the loss of spatial information [8,9,10,11,12].

Laser capture microdissection-mediated RNA sequencing (LCM-seq) is an alternative/complementary spatial transcriptomics method to the methods described above [13,14,15,16]. This technique is relatively inexpensive and requires less time and effort relative to most other spatial transcriptomic techniques. It was first devised by researchers at the National Institute of Health [17] and has observed widespread use in the biological sciences. Due to the small area where the laser beam can be focused, microdissection can be performed in regions as small as a single cell, or even at sub-cellular domains, such as at the chromosome level [18,19]. Laser microdissection can be used to study genomics [18,20], transcriptomics [18,20,21], and proteomics [20,22,23] of a wide array of tissue types. The technique has been extensively used in different diagnostic laboratories for the detection of abnormal tissues such as during cancer [24,25] and other pathological abnormalities [26,27], brain anomalies connected to animal behavior [28], in the identification of neuronal cell types [29], and in basic and applied biological research such as in developmental biology [30], forensic research [31], and plant sciences [32].

The technique involves fixing biological tissues using a non-crosslinking fixative such as alcohol and/or acetone [33] on a membrane slide made with polyethylene naphthalate (PEN) or polyethylene terephthalate (PET) and using an ultraviolet (UV) or infrared (IR) laser beam to melt and dissect the membrane along with the tissue [17,34]. The microdissected tissue is captured either on a thermolabile adhesive film that, upon melting with an IR laser, sticks to the surface of the target tissue [17,18], or on adhesive caps at the top of the membrane slides, propelled there by the upward force applied by the UV laser [18].

Here, we describe a laser-capture microdissection protocol for developing butterfly wing tissue. We show how to fix and stain the butterfly wing tissue, perform the laser microdissection, isolate the microsections using fine tweezers, extract RNA from the isolated tissue, and perform RNA quality analysis. The butterfly wings because of their thickness (above 40 µm) are difficult to capture using the upward force of UV lasers and hence we used fine tweezers to isolate the microtissues from the dissected samples. We successfully applied this technique for the isolation of RNAs from the intervein sectors of larval wing tissues, from the larval eyespot centers, and from the pupal eyespot rings with RIN (RNA integrity number) values above 8.0. This technique will be helpful to other researchers who are interested in similar microsections of their tissues, who are working on similar thick tissue types, and who have difficulty in the extraction of high-quality RNA from PEN membrane-bound and ethanol-/acetone-fixed tissues.

## 2. Experimental Design

The major steps involved in the microdissection and isolation of RNA from laser micro-dissected wing tissues include: (1) Tissue fixation using ethanol and acetone and transfer onto a PEN membrane slide; (2) Wing staining using a cell staining solution—this allows better visualization of transparent tissues; (3) Wing microdissections using a laser microbeam microscope; (4) Isolation of the micro-dissected samples using fine tweezers; and (5) Extraction of RNA from the microsections.

Table 1 shows the details involved in each step and Figure 1 illustrates the workflow of the whole procedure.

## 3. Required Materials and Equipment

### 3.1. Materials

Microcentrifuge tubes 1.5 mL (Eppendorf, Hamburg, Germany; Cat. No. T9661-500EA)PCR tubes 200 µL (Axygen, Union City, AZ, USA; Cat. No. 14-222-262)PEN membrane slides (PEN Membrane Glass Slides; Thermo Fisher Scientific, Waltham, MA, USA; Cat No. LCM0522)Fine straight tweezers (Dumont, Montignez, Switzerland; Cat. No. 11254-20)Stainless steel beads, 0.5 mm (Next Advance, Troy, NY, USA; Cat. No. SKU: SSB05)Filter Pipette tips (Axygen 200 µL and 1000 µL Universal Fit Filter Tips; Corning, AZ, USA; Cat. No. TF-200-R-S, TF-1000-R-S)Ice bucket with closed lid (Coleman; 5L)Flat spatula (Thomas Scientific; Thomas Scientific, Swedesboro, NJ, USA; Cat. No. 1208Y75)Zirconium beads, 0.1 mm (BioSpec; Bartlesville, OK, USA, Cat. No. 11079101z)LCM Adhesive Cap 500 (Zeiss, Oberkochen, Germany; Cat. No. 415190-9201-000)

### 3.2. Reagents for Fixation and Staining the Wing Tissue

Histogene staining solution (Thermo Fisher Scientific, Waltham, MA, USA; Cat. No. KIT0415)Ethanol (Sigma Aldrich (Merck), Burlington, MA, USA; Cat. No. E7023-1L)Acetone (Sigma Aldrich (Merck), Burlington, MA, USA; Cat. No. 179124-1L)RNaseZap RNase Decontamination Solution (Thermo Fisher Scientific, Waltham, MA, USA; Cat. No. AM9780)UltraPure DNase/RNase-Free Distilled Water (Thermo Fisher Scientific, Waltham, MA, USA; Cat. No. 10977015)Proteinase K (NEB, Ipswich, NY, USA; Cat. No. P8107S)

### 3.3. Reagents for RNA Isolation and Verification of RNA Integrity

Qiagen RNA isolation kit (RNeasy Plus Mini Kit (250); Qiagen, Hilden, Germany.; Cat. No. 74136)Agarose molecular grade (Vivantis; Cat. No. PC0701-500g)Trizma base (Sigma-Aldrich (Merck), Burlington, MA, USA; Cat. No. T1503-500G)EDTA (Thermo Fisher Scientific, MA, USA; Cat. No. 17892)Sodium Dodecyl Sulfate (SDS; Sigma-Aldrich (Merck), Burlington, MA, USA; Cat. No. 151-21-3)SYBRSafe (Invitrogen, Carlsbad, CA, USA; Cat. No. S33102)DNA ladder (ExactMark 100bp DNA Ladder; Axil Scientific, Singapore; Cat. No. BIO-5130-100ug)RNA Gel Loading Dye (Thermo Fisher Scientific, Waltham, MA, USA; Cat. No. R0641)

### 3.4. Equipment

Zeiss Stemi 305 stereo microscope (Zeiss, Oberkochen, Germany)Zeiss laser microdissection microscope (Zeiss PALM microbeam-Laser microdissection; Zeiss, Oberkochen, Germany)Gel electrophoresis system (Bio-Rad, Hercules, CA, USA)Gel documentation system (Azure 200 Gel Imaging Workstation; Azure Biosystems, Dublin, CA, USA)Nanodrop ND-1000 Spectrophotometer (Nanodrop ND-1000; Thermo Fisher Scientific, Waltham, MA, USA)Homogenizer (Next Advance Bullet Blender; Next Advance, Troy, NY, USA)

## 4. Procedure

### 4.1. Fixation of Wing Tissue Using Ethanol and Acetone on a PEN Membrane Slide

Dissect the butterfly larval and pupal wings based on the protocol described in [35] and transfer wings into a 1.5 mL microcentrifuge tube with 1X PBS.Dehydrate the wings by adding a gradual concentration of 50, 75, and 100% ice-chilled alcohol mixed with PBS. Alternatively, dissected wings can be transferred to a PEN membrane slide and dehydrated on the slide by adding a gradual concentration of 50, 75, and 100% ice-chilled alcohol.


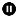
  **PAUSE STEP** Dehydrated wings can be stored at −80 °C for up to 6 months.

3.After dehydration, move the wings to a PEN membrane slide. For the larval wings, use a 1000 µL pipette to transfer the wings and for the pupal wings use a flat spatula. The wings once dry will stick to the membrane of the slide.4.Fix the wings twice with 100% ice-chilled acetone using a 1000 µL pipette.


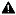
  **CRITICAL STEP** Keep the PEN membrane slide in an ice box as much as possible to avoid RNA degradation.

### 4.2. Staining of Wing Tissue

After wing fixation use Histogene staining solution to stain the wings. Gently add a few drops of the staining solution on top of the wings and leave for 10–40 s, until a purple color is seen on the wing.Wash the wings with 100% EtOH using a 1000 µl pipette three times and let the wings dry on the slide.Move the slide to a 50 mL falcon tube.


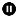
  **PAUSE STEP** Wings can be stored inside the falcon tube for over 6 months in a −80 °C freezer.

### 4.3. Dissection of the Wing Tissue Using a Laser Microdissection Microscope

Move the slide inside an ice box (Coleman ice bucket with crushed ice) to the laser microdissection microscope room.Turn on the microscope, turn on the PALMRobo 4.8 pro software, and mount the slide for dissection (Figure 2A,B).To mount the slide and the adhesive cap (LCM Adhesive caps 500), click on the button indicated by the red and yellow arrow in PALMRobo 4.8 pro software indicated in Figure 2C.Note: The adhesive caps are used only to allow the microscope to operate. The butterfly wing microsections are thick and hence very difficult to capture using the adhesive caps.Use the settings as mentioned in Table 2 for optimal cutting of the butterfly wing tissue.Use the tools highlighted inside the red box in Figure 2C to draw the microsections of interest and perform the tissue dissection by clicking the cut button (black arrow in Figure 2C).After the microdissection, transfer the slides back to the ice box and bring back to lab for isolation of the microsections and extraction of RNA.


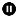
  **PAUSE STEP** Wings can be stored at this stage at −80 °C for up to 6 months.

### 4.4. Isolation of the Microdissected Tissues Using Fine Tweezers

**1.** Move the PEN membrane slide with the microdissected samples on an ice-filled Petri plate under a normal dissection microscope (Zeiss Stemi 305).**2.** Prepare 200 µL tubes with 20 µL of molecular grade water on ice for the transfer of the microsections.**3.** Using a fine tweezer carefully pick up the microsections and move them to the respective 200 µL tubes.**4.** After all the sections are collected, move to RNA isolation.


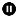
  **PAUSE STEP** Wings can be stored at this stage at −80 °C for up to 6 months.

### 4.5. RNA Extraction from the Microdissected Samples

Add 100 µL of LCM lysis buffer (1% SDS in Tris-EDTA) and 10 µL of proteinase K to the 200 µL tubes with the microsections.Incubate the tubes at 55 °C in a water bath, vortex every 3 min, for a total of 9 min. At this step the microsections will partly dissolve in the LCM lysis buffer.After this step, proceed to RNA extraction using a kit of choice.

#### 4.5.1. RNA Extraction Using Qiagen RNeasy Plus Mini Kit

Transfer the 130 µL solution from the previous step to a 1.5 mL microcentrifuge tube.Add 350 µL of RLT plus buffer and a small amount of 0.1 mm zirconium beads.Place the tube in a homogenizer and homogenize for 3 min at the maximum rate.Remove the tubes and centrifuge at 14,000 rpm at 4 °C for 3 min.Transfer the supernatant to a gDNA eliminator spin column and centrifuge at 14,000 rpm for 30 s.Discard the column and to the flow-through add 350 µL 70% EtOH. Mix well and transfer to RNeasy spin column.Centrifuge at 14,000 rpm for 30 s and discard the flow-through.Add 700 µL of RW1 buffer to the column and centrifuge at 14,000 rpm for 30 s. Discard the flow-through.Add 500 µL of RPE buffer to the column and centrifuge at 14,000 rpm for 30 s. Discard the flow-through.Add 500 µL of RPE buffer to the column and centrifuge at 14,000 rpm for 2 min. Discard the flow-through and transfer the column to a 1.5 mL microcentrifuge tube.Add 20 µL molecular grade water to the column and let it sit for 3 min before centrifugation.Centrifuge at 14,000 rpm for 1 min. Discard the column and measure the concentration of the RNA in a nanodrop machine.

#### 4.5.2. Evaluation of the Extracted RNA Using Agarose Gel Electrophoresis

Prepare a 2% agarose gel by mixing 1 gm of agarose to 50 mL 1X TAE buffer.Mix 3 µL of RNA loading dye and 3 µL of RNA sample and load it in one of the agarose gel wells. Add ladder in the adjacent well.Run the gel for 30 min at 120 V and 400 mA.Image the gel in a gel documentation system. The RNA should appear as a clear band in the gel.

## 5. Expected Results

### Staining, Dissection, and Analysis of Extracted RNA

After Histogene dye staining, certain regions of the wing with denser cell populations will be visible. These include the eyespot centers, the intervein cells, and the wing margin (Figure 3B,C,E,F,H,I). Reference images from a comparable developmental stage stained for particular mRNAs using a technique such as hybridization chain reaction (HCR) [36] or for particular proteins [37,38] can also be used to localize the target cell population (Figure 3A,D,G). The dissection will result in clear visible marks of the dissected tissue which can be removed easily using a pair of fine tweezers (Figure 3J).

After the RNA extraction, the RNA sample appears as a clear band in an agarose gel (Figure 3K) and has an RIN value above 8.0 (Figure 3L).

The concentration of RNA from 30 larval wings is usually above 1000 ng for tissue from each wing sector (described in Figure 4) and eyespots (Table 3). For the larger pupal wings, 4–5 wings are sufficient for a total of 1000 ng of RNA from the eyespots (Table 3).

## 6. Discussion

Laser microdissection of tissues has remained a popular tool to address multiple questions related to pattern formation and developmental biology, and to characterize pathological abnormalities within tissues. Traditional LCM protocols involve paraffin embedding and/or the use of a microtome (cryostat) for producing thin slices of tissues which can then be used for capturing tissue sections using an LCM microscope [32,34,39]. These methods require additional materials and instruments that sometimes affect the quality of the RNA. In the present protocol, we have isolated microsections of thick wing tissue using a pair of fine tweezers, which are difficult to capture using the traditional LCM method. Using 10% SDS solution in Tris-EDTA (LCM lysis buffer), and proteinase K along with a heating step at 55 °C, we showed that the tissues fixed to the PEN membrane slides could be efficiently dissolved in the buffer. This step, followed by homogenization with microbeads, allows for high yield of RNA with a good RIN value for downstream transcriptomic applications. This protocol can extract microsections of tissue as small as approximately 10,000 µm^2^, after which it becomes difficult to isolate the microsections using tweezers. Hence, the protocol is not suitable for cellular or subcellular LCM applications. Furthermore, the present method cannot be applied to much thicker tissues from organs such as the brain, liver, etc., for which a microtome is required.

The integration of LCM data with that from single cell, multiplex FISH, and spatial sequencing will help address the limitations of each technique, to produce more robust spatial transcriptomes.

## 7. Reagents Setup

### 7.1. LCM Lysis Buffer (Table 4)

In a 50 mL sterile falcon tube add 0.5 mL of 1M Tris Hcl (pH 8.0) and 0.1 mL 0.5M EDTA (pH 8.0).Prepare 10% SDS by mixing 10 g of SDS in 90 mL of dH_2_O in a beaker. Mix with gentle stirring.Add 5 mL of the 10% SDS and dH_2_O to the falcon tube till 50 mL and mix properly by shaking.

### 7.2. For 50X TAE Buffer (Table 5)

Add 242 g of Trizma base and 18.61 g of EDTA in a 1-L bottle.Add 500 mL dH_2_O and add 57.1 mL glacial acetic acid.Adjust the volume to 1 L and autoclave.

## Figures and Tables

**Figure 1 mps-05-00067-f001:**
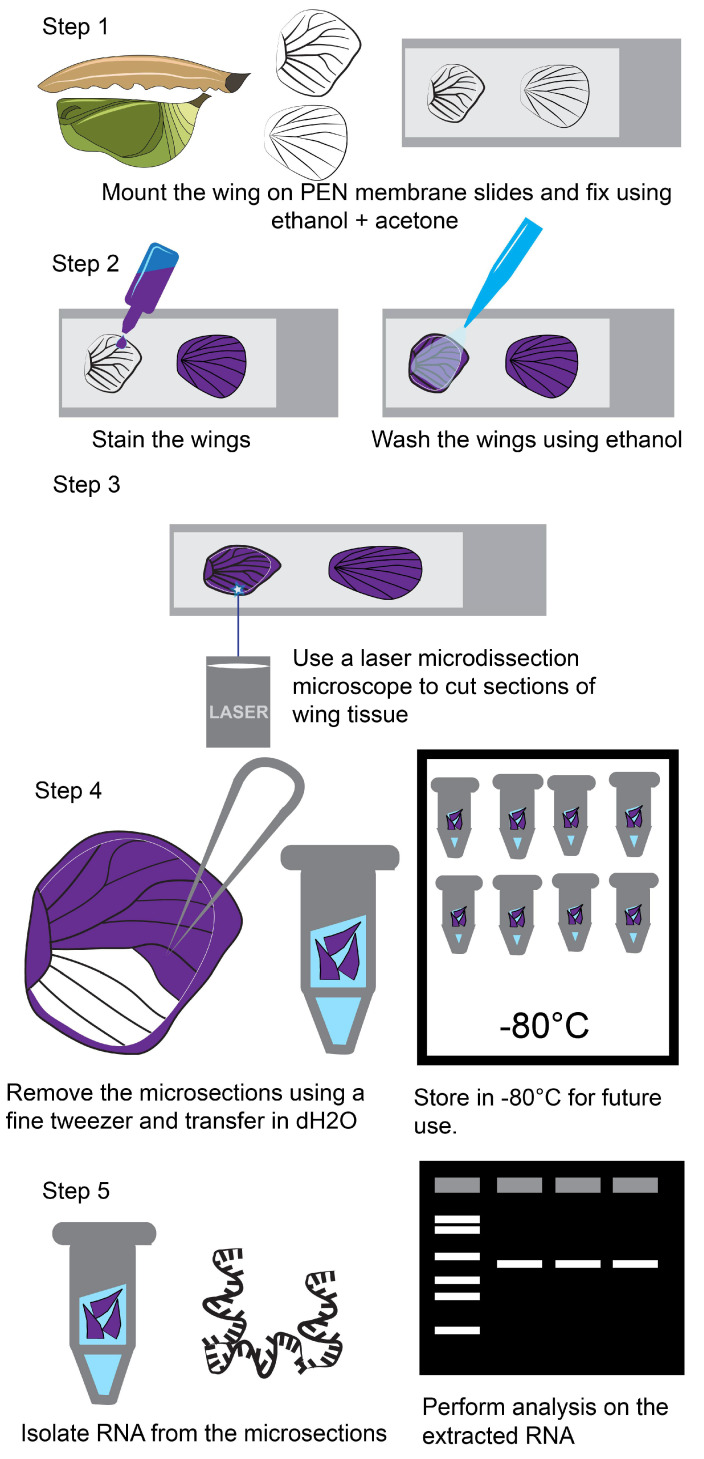
**Steps involved in the microdissection of butterfly larval and pupal wing tissue.** Step 1 involves the transfer of the wing tissue onto the PEN membrane slides and fixation using ethanol and acetone. Step 2 involves staining the wings with Histogene solution and performing washes with ethanol. Step 3 involves laser microdissection using a Zeiss PALM microbeam. Step 4 involves isolation of the microsections from the dissected microtissues using a pair of fine tweezers and transfer of microsections into molecular grade water. Afterwards, the tissues can be stored at −80 °C. Step 5 involves extraction of RNA from the microdissected tissues and testing the integrity of the RNA.

**Figure 2 mps-05-00067-f002:**
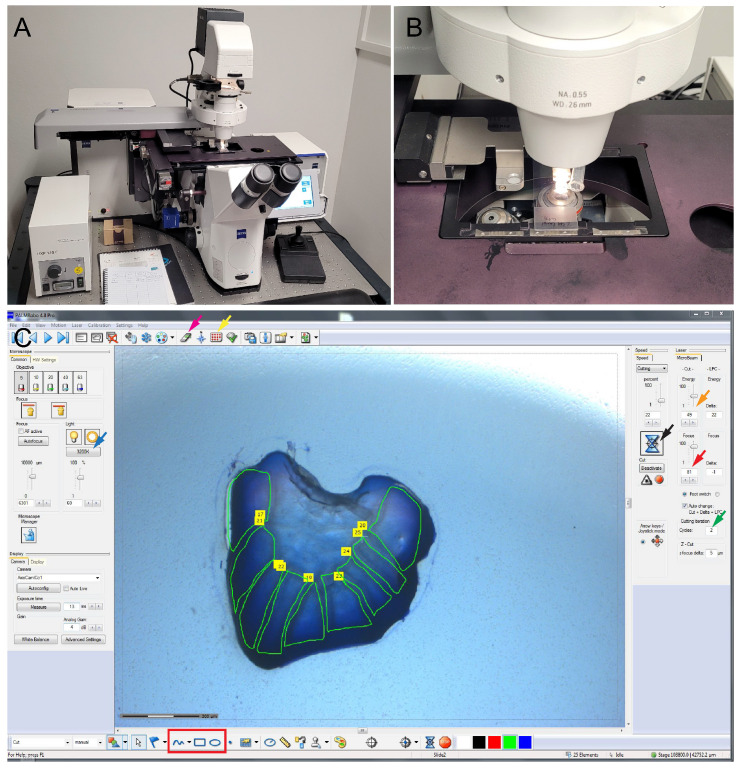
**Zeiss PALM microbeam-laser microdissection microscope and PALMRobo 4.8 Pro settings for laser microdissection of *Bicyclus anynana* larval and pupal wings**. (**A**) Zeiss PALM microbeam-laser microdissection microscope. (**B**) A closeup of the mounted slide on the microscope. (**C**) Settings used for laser microdissection of the larval and pupal wing tissues. Magenta arrow: slide holder control; yellow arrow: collection cap control; orange arrow: laser energy; red arrow: laser focus; green arrow: number of laser cutting cycles; blue arrow: intensity for the backlight; and red box: tools for selecting the sections for microdissection.

**Figure 3 mps-05-00067-f003:**
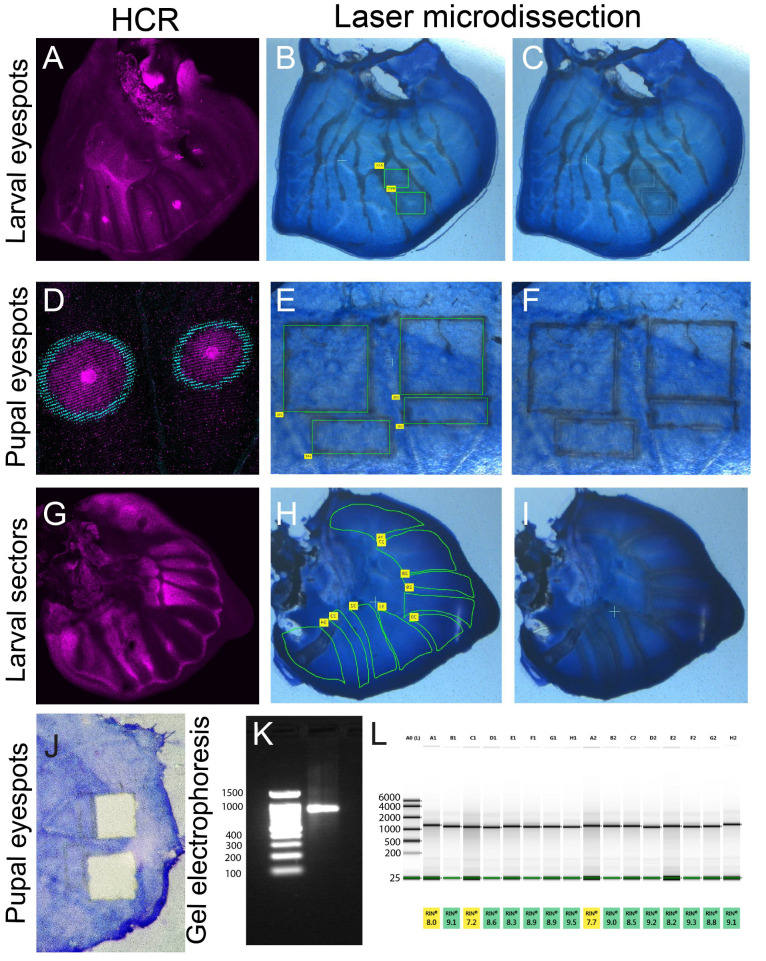
**Laser microdissection of larval and pupal wings.** (**A**) Larval wings showing the expression of the gene *spalt* in the center of the eyespots (visualized with a hybridization chain reaction). (**B**,**C**) Panel A is used as a reference to localize the position of the eyespot center in a wing at a similar developmental stage (**B**,**C**) and to perform the microdissections. (**D**) Pupal wing, expressing the genes *spalt* (magenta) and *optix* (cyan) in the rings of the eyespots (hybridization chain reaction) is used as a reference to localize the position of the eyespots in a wing at a similar developmental stage (**E**,**F**) to perform microdissections. (**G**) Larval wing showing expression of the gene *spalt* in the different sectors of the wings (hybridization chain reaction) is used as a reference to localize the positions of the wing sectors along with the venation pattern on a wing at a similar developmental stage (**H**,**I**) to perform microdissections for sector-specific transcriptomes. (**J**) A pupal wing with the eyespot regions removed with a pair of fine tweezers. (**K**) An agarose gel electrophoresis on the RNA extracted from laser-dissected wing tissue with a clear band of RNA. (**L**) RIN value of different RNA samples from the larval eyespot and control regions from wet season (WS) and dry season (DS) forms of *B. anynana* butterflies. A0: ladder; A1, C1, E1, G1: WS control; A2, C2, E2, G2: DS control; B1, D1, F1, H1: WS eyespot; B2, D2, F2, H2: DS eyespot. Green line: lower marker. The samples were sent for RNA-seq and the gel image was generated from Azenta, Singapore.

**Figure 4 mps-05-00067-f004:**
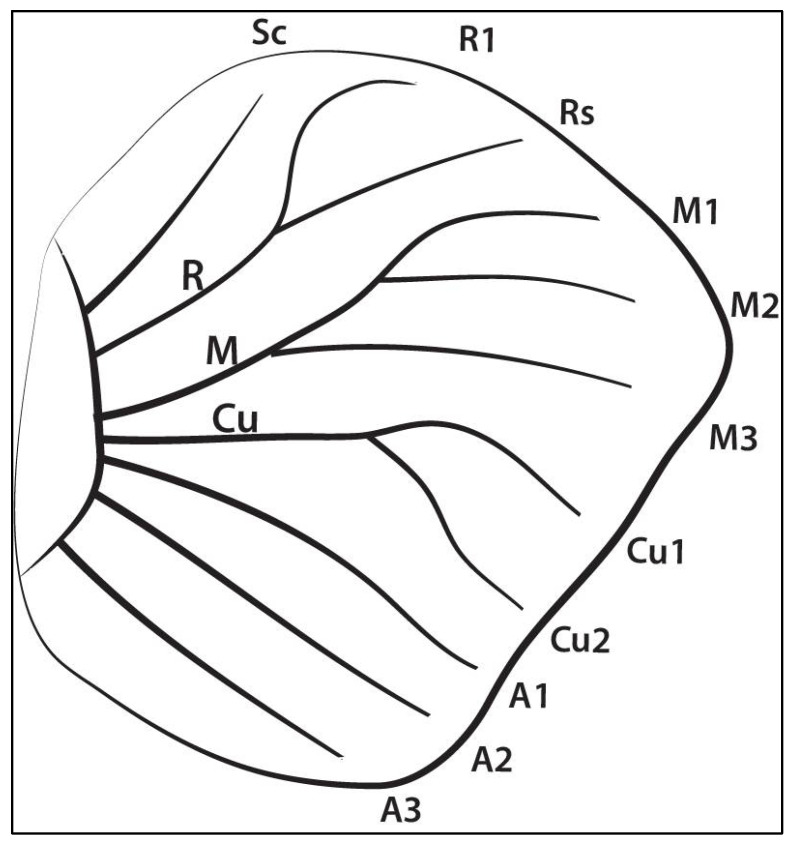
Venation nomenclature of *Bicyclus anynana* larval hindwing used for dissection and isolation of sector-specific transcriptomes using LCM.

**Table 1 mps-05-00067-t001:** Experimental stages and time needed to complete each stage.

Experimental Stages	Time for Completion
Fixation of tissue using ethanol and acetone on a PEN membrane slide.	10 min
2. Staining the wings using Histogene staining dye.	10 min
3. Microdissection using Zeiss PALM microbeam laser microscope.	1–3 h
4. Isolation of the microdissected samples using fine tweezers.	1–2 h
5. Isolation of RNA from the microsections of the wing tissue and checking the integrity of RNA.	2 h

**Table 2 mps-05-00067-t002:** Settings for the PALMRobo 4.8 pro software for the dissection of butterfly larval and pupal wing tissues.

Sl. No.	Name	Setting
1	Light	3200 K
2	Cut energy	50%
3	Focus	81
4	Cutting iteration	2
5	Z-focus delta	5 µm

**Table 3 mps-05-00067-t003:** RNA concentration obtained after pooling microsamples from different wings. The sector description is provided in Figure 4.

Sl. No.	Sample Name	No. of Wings Used	RNA Amount (ng) *
1	Sector 1 (Rs–anterior margin)	30	1596
2	Sector 2 (M1-Sc+Rs)	30	1406
3	Sector 3 (M2-M1)	30	1197
4	Sector 4 (M3-M2)	30	760
5	Sector 5 (Cu1-M3)	30	1026
6	Sector 6 (Cu2-Cu1)	30	1520
7	Sector 7 (A1-Cu2)	30	1178
8	Sector 8 (A2-A1)	30	1007
9	Sector 9 (A2–posterior margin)	30	2109
10	Pupal eyespot	4	1843
11	Pupal control	4	1197
12	Larval eyespot (WS)	30	1638
13	Larval control (WS)	30	650

*: The RNA amount values for larval eyespot and control tissues were averaged between four biological replicates. Since comparative amounts of total RNAs were obtained from both DS and WS samples, only WS samples are shown.

**Table 4 mps-05-00067-t004:** Chemicals for LCM lysis buffer.

Sl. No.	Chemical	Amount
1	1M Tris HCl (pH 8.0)	0.5 mL
2	0.5M EDTA (pH 8.0)	0.1 mL
3	10% SDS	5 mL
4	dH_2_O	Till 50 mL

**Table 5 mps-05-00067-t005:** Chemicals for 50× TAE buffer.

Sl. No.	Chemical	Amount
1	Trizma base	242 g
2	EDTA	18.61 g
3	Glacial Acetic Acid	57.1 mL
4	dH_2_O	Till 100 mL

## Data Availability

Not applicable.

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
