# Peer review of "Laser Microdissection-Mediated Isolation of Butterfly Wing Tissue for Spatial Transcriptomics"

_mps, 2022, doi:10.3390/mps5040067_

Round 1

Reviewer 1 Report

The authors describe a laser-dissection protocol for total RNA extraction from butterfly larval and pupal wing tissues that can be useful to researchers interested in the transcriptome of specific areas of the wing during development. It partly solved the assignment problem of specific patterns of gene expression to specific cells in a complex tissue. There are several major problems:

1.      Since there are many analyses, the authors should provide the data, images, and software or scripts onto GitHub or other publicly available websites.

2.      The authors used butterfly wing tissue as an example. Is there uniqueness of this tissue? If we work on other tissues, which perspectives do we need to pay attention to? The authors should discuss this.

3.      The authors should compare this protocol with other protocols and show the improvements and discuss why the different procedures can make such improvements.

4.      Although it is a protocol, the authors still need to show the novelty and their contribution to this field. Please highlight such novelty or contribution in the abstract. The current abstract is descriptive and has no collusion.

Author Response

The authors describe a laser-dissection protocol for total RNA extraction from butterfly larval and pupal wing tissues that can be useful to researchers interested in the transcriptome of specific areas of the wing during development. It partly solved the assignment problem of specific patterns of gene expression to specific cells in a complex tissue. There are several major problems:

  1. Since there are many analyses, the authors should provide the data, images, and software or scripts onto GitHub or other publicly available websites.

Answer: Thank you for taking time to review our manuscript. The present manuscript doesn’t include any software developed by us. The software mentioned in the protocol is from Zeiss which is proprietary and hence cannot be uploaded to a public website. We have added all the relevant images in the manuscript. If any additional information is required for a particular section kindly let us know.

  1. The authors used butterfly wing tissue as an example. Is there uniqueness of this tissue? If we work on other tissues, which perspectives do we need to pay attention to? The authors should discuss this.

Answer: The butterfly wing tissue during the larval stage has two layers of epidermis covered by a peripodial membrane that adds to the thickness of the tissue. This thick tissue cannot be captured using the traditional LCM method and hence, after cutting two layers of the tissue with z-delta 10 µm, we have used fine tweezers to isolate the sections from the wing. We have mentioned this in the protocol. This method can be used for similar thick tissues.

  1. The authors should compare this protocol with other protocols and show the improvements and discuss why the different procedures can make such improvements.

Answer: Thank you for this suggestion. We have made modifications to the manuscript to address the main improvements of our protocol, relative to pre-existent protocols, in the discussion section.

  1. Although it is a protocol, the authors still need to show the novelty and their contribution to this field. Please highlight such novelty or contribution in the abstract. The current abstract is descriptive and has no collusion.

Answer: We have modified the abstract to highlight the novelty of the present protocol.

Reviewer 2 Report

The authors present a well-written protocol that describes the steps for LCM and RNA extraction of butterfly wings. Overall, a nice description and it is well-written.

Unfortunately, the title is misleading as transcriptomics is not including in the protocol and description. The authors should revise the title and abstract or add sections for library building, pooling, sequencing and analysis (strongly recommend).

The article could use a better discussion of challenges, key steps and analysis of RNA quality, and as mentioned above, it currently lacks any description of transcriptome analysis for Bicyclus anynana.

Minor point, the molecular weight markers sizes in figures should be labeled (i.e., Fig3)

Author Response

The authors present a well-written protocol that describes the steps for LCM and RNA extraction of butterfly wings. Overall, a nice description and it is well-written.

Unfortunately, the title is misleading as transcriptomics is not including in the protocol and description. The authors should revise the title and abstract or add sections for library building, pooling, sequencing and analysis (strongly recommend).

Answer: Thank you for taking the time to review our manuscript. We have changed the title of the manuscript. After isolation, we send the RNA to a company (Azenta Life Sciences) to carry out library building, pooling, and sequencing.

The article could use a better discussion of challenges, key steps, and analysis of RNA quality, and as mentioned above, it currently lacks any description of transcriptome analysis for Bicyclus anynana.

Answer: We have now added a discussion section on the challenges, and key steps in our protocol.

Minor point, the molecular weight markers sizes in figures should be labeled (i.e., Fig3)

Answer: We have added the molecular weight marker sizes.

Round 2

Reviewer 1 Report

The authors have addressed my questions.

Reviewer 2 Report

The revisions are satisfactory.